# BALTO: FAST TENSOR PROGRAM OPTIMIZATION WITH BIASED-DIVERSITY-BASED ACTIVE LEARNING

**Jun Bi[1,2,3], Xiaqing Li[2], Qi Guo[2], Rui Zhang[2], Yuanbo Wen[2], Xing Hu[2], Zidong Du[2], Xinkai Song[2], Yifan Hao[2], Yunji Chen[2]\***
[1]University of Science and Technology of China
[2]SKL of Processors, ICT, CAS
[3]Cambricon Technologies, Beijing, China

## ABSTRACT

Tensor program optimization (TPO) based on pre-trained models can effectively reduce the computing time of deep neural networks. However, training of such models is prohibitively expensive, which highly depends on a large-scale dataset and thus requires tremendous time-consuming performance measurements (more than 1 million) on target platforms. In this paper, we propose BALTO, a fast TPO approach with biased-diversity-based active learning, aiming at significantly reducing training costs under similar program performance optimization ability. The key insight is that random sampling of existing approaches suffers from a heavy redundancy of low-performance programs, which incurs tremendous time-consuming measurements. Inspired by this, BALTO removes such redundancy by introducing active learning (AL) to TPO for a much lower training cost. However, applying AL with a brute-force way in BALTO can lead to an overestimation problem. To address this, we further propose a biased-diversity-based diversity scheme specially designed for BALTO. We compare BALTO against TenSet on 6 typical hardware platforms over 2 learning models. Experimental results show that, on average, BALTO only requires 5% of the total measurements of TenSet to achieve the same or higher model accuracy. Moreover, the optimized tensor programs even outperform that of TenSet by 1.07% due to higher model accuracy.

## 1 INTRODUCTION

Tensor program optimization (TPO) can effectively reduce the computing time of neural networks by searching for high-performance programs in a designed search space (Chen et al., 2018; Zheng et al., 2020a;b). In TPO, neural networks are first represented as tensor programs that describe the computation of multi-dimensional data arrays. Then performances of these tensor programs are measured on a target hardware platform. Such measurements are time-consuming and thus optimizing a given network can cost several days or even weeks, which greatly hinders the wide application of TPO (Zhu et al., 2022). To accelerate TPO, pre-trained machine learning models (Adams et al., 2019; Haj-Ali et al., 2020; Anderson et al., 2020; Zheng et al., 2021) are introduced to replace a substantial part of the hardware measurements with performance predictions of a pre-trained model. For example, as the state-of-the-art approach, TenSet (Zheng et al., 2021) can significantly reduce the optimization time by up to $10\times$ through training on a large-scale and well-established dataset.

However, training the models is prohibitively expensive. The main reason is that these models highly depend on a large-scale training dataset. Unfortunately, collecting such a dataset involves massive performance measurements on hardware platforms, suffering from excessively long execution time. For example, for each hardware platform, around 8 million different tensor programs are required to be measured (Adams et al., 2019; Zheng et al., 2021). Even on a high-end GPU like V100, such measurements still can consume about 4000 GPU hours. This burden could be much worse in those capability-limited systems, e.g., mobile devices such as NVIDIA Jetson TX2, most of which often require more than 5000 GPU hours to conduct the measurements. More importantly, a much larger dataset is required in real-world optimization tasks, so as to achieve better model generalization

---

\*Corresponding author.

on different tensor programs. Consequently, as the size of the dataset increases, the number of the measurements can be significantly increased correspondingly. This can lead to great consumption of time and energy, and thus hinders wide deployment of ML-based TPO in industrial practice.

To look more closely at the inside of the large-scale datasets sampled randomly by existing approaches, we conduct a deep exploration of the distribution of these datasets. We observe that random sampling adopted in existing approaches can result in imbalanced training datasets, where the high-performance programs are excessively less than that of the low-performance ones. We randomly sample 90,000 tensor programs of TenSet (as shown in Figure 1) and find that high-performance programs only account for 19% and 8% of the to-

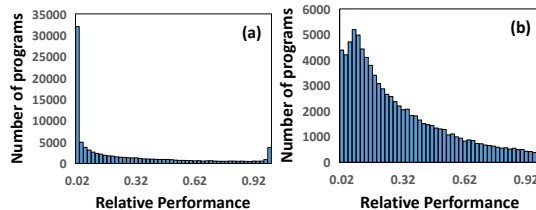

Figure 1: The ratio of high-performance programs is much smaller than low-performance programs. (a) NVIDIA T4. (b) Platinum 8272CL.

tal dataset on Platinum 8272CL CPU and T4 GPU respectively. In contrast, the low-performance programs take the bulk of the total dataset (81% and 92% respectively). Such imbalance can further lead to a heavy redundancy of the low-performance programs that incurs tremendous time-consuming measurements. The main reason behind the redundancy is that the importance of generated tensor programs is highly different. In fact, program optimization pays more attention to those high-performance programs. Excessively more low-performance programs cannot offer additional benefit of predicting high-performance programs, thus being a heavy redundancy.

To this end, we propose BALTO, a fast TPO approach with biased-diversity-based diversity active learning, aiming at significantly reducing training costs under similar optimization ability. The key insight is that random sampling of existing approaches suffers from a heavy redundancy of low-performance programs, which incurs tremendous time-consuming measurements. Inspired by this, BALTO removes such redundancy by introducing active learning (AL) to TPO for a much lower training cost. In this way, the measurements can be significantly lowered, thus greatly reducing the training cost of building pre-trained model. However, applying AL in a brute-force way in BALTO can lead to an overestimation problem, where the relative performance of the estimated accuracy is much better than that of the ground truth. To address this problem, we further propose a biased-diversity-based scheme specifically designed for BALTO, which can efficiently reduce the distribution imbalance of the sampled programs caused by overestimation.

Finally, we integrate BALTO into TenSet, and compare BALTO with state-of-the-art baseline TenSet on 6 typical hardware platforms (i.e., two GPU platforms and four CPU platforms) over two learning models (i.e., XGBoost, MLP). The experimental results show that BALTO achieves same or higher model accuracy while only requiring 5% of TenSet's hardware measurements. Moreover, the optimized tensor programs even outperform that of TenSet by 1.07% due to higher model accuracy.

To the best of our knowledge, BALTO is the first work to reduce the training cost of pre-trained-model-based TPO by introducing AL. Summarily, our key contributions are three-fold:

- We conduct a deep exploration on the distribution of large-scale datasets sampled randomly by existing approaches, and observe that the random sampling results in an imbalanced dataset, and thus suffers from heavy redundancy in the training dataset.

- We propose BALTO, a fast TPO approach with active learning, aiming at significantly reducing training cost under similar optimization ability. To address the overestimation problem, we further propose a biased-diversity-based scheme specially deigned for BALTO.

- We conduct a comprehensive performance evaluation on six typical hardware platforms, indicating that BALTO achieves same or higher model accuracy with $20\times$ reduction in performance measurements. Moreover, the optimized programs even outperform that of TenSet by 1.07% on NVIDIA T4.

## 2 BACKGROUND

### 2.1 TENSOR PROGRAM OPTIMIZATION.

TPO is a process of transforming an input tensor program into another tensor program with optimal or near-optimal performance. Figure 2 demonstrates a workflow of a typical TPO. As shown in Figure 2, it mainly consists of three modules, including program performance estimation, optimization space exploration, and program performance evaluation. The program performance estimation module is used for predicting the performance of input programs, which is usually composed of machine learning models or expert-designed models. The optimization space exploration module searches for high-performance program transformations in a pre-defined search space and outputs a batch of programs with high scores on predicted performance. The program performance evaluation module compiles the batch of programs and measures their execu-

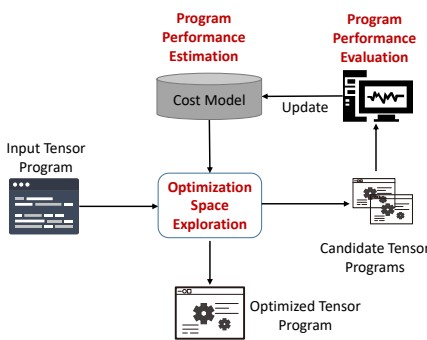

Figure 2: Workflow of TPO.

tion times on the hardware platform. The measured performance can be used for further fine-tuning the models. Finally, the program with the highest measured performance is output as the optimization result.

## 2.2 PRE-TRAINING OF PERFORMANCE MODEL.

Pre-training of the performance model has two key stages typically: the sampling stage and the training stage. In the sampling stage, model pre-training usually depends on a large-scale dataset based on program performance, which is generated by performance measurements on target hardware platforms. Concretely, the sampling process mainly consists of three steps. First, a set of program optimization tasks are generated by the compiler, such as conv2d operator with different shapes. Second, for each generated task, random sampling is conducted in the optimization search space to produce a set of tensor programs without performance measurement. Third, these tensor programs are delivered to the module of performance evaluation for performance measurement, and finally, the measured performance and the features of tensor programs are kept in the dataset.

In the training stage, generally, the performance model aims at fitting the relative performance of the programs, which can be denoted as $y = \frac{T}{T_{max}}$. Concretely, $T$ denotes the throughput of the program, and $T_{max}$ denotes the maximum throughput that can be reached by the corresponding optimization task of that program. Since the maximum throughput $T_{max}$ of the task is unknown, $T_{max}$ usually can be estimated by the maximum throughput of the sampled programs. Importantly, such estimation can be quite accurate if given a large number of samples. The commonly-used training models include XGBoost, MLP, and LSTM, etc, and the goal of model training is to minimize $rmse$ or $lambdarankloss$. The evaluation criteria of the model is greatly critical. Compared with $rmse$ or $R^2$, the Top-k performance predicted by the model on each optimization task is more related to the optimization quality of the compiler (Zheng et al., 2021). For a given set of tensor programs, the Top-k performance can be calculated as the division of the predicted Top-k programs' average performance and the optimal programs' performance.

## 2.3 DIVERSITY-BASED ACTIVE LEARNING

The training of machine learning, especially deep learning, highly depends on a large number of labeled samples whose generation can be prohibitively expensive. Active learning (AL) is a popular training framework, aiming at effectively reducing the requirement of expensive labels. In each iteration of an AL process, unlabeled samples are firstly selected from a pool with a fixed size, then the selected samples are delivered to be labeled, and finally these labeled

---

**Algorithm 1** Core-set Greedy Selection

**Require:** $D_l$, $D_u$, $B_t$
**Output:** $S_t$ is a set of samples to be labeled
1: $S_t \leftarrow \phi$
2: **while** $|S_t| < B_t$ **do**
3: $\quad u = \arg\max_{x_i \in D_u} \min_{x_j \in D_l \cup S} d(x_i, x_j)$
4: $\quad S_t \leftarrow S_t \cup \{u\}$
5: **return** $S_t$

---

samples are used for model training. The core of AL is the selection method which can effectively reduce the labeling redundancy while achieving similar model accuracy. A commonly used ap-

proach for selection is diversity-based selection, which selects representative samples for labeling to increase diversity.

The Core-set selection (Sener & Savarese, 2018), one of the state-of-the-art diversity-based approaches, formulates sample selection as solving the *k-center* optimization problem of $\min_{S_t \subseteq D_u} \max_{x_i \in S_t} \min_{x_j \in D_l \cup S_t} d(x_i, x_j)$ *s.t.* $|S_t| \leq B_t$, where $S_t$ is the samples selected at time step $t$, $d(x_i, x_j)$ is a measure of the distance between sample $x_i$ and $x_j$, and $B_t$ is the total budget of samples at step $t$. Since the problem is NP-Hard, a common approach leverages the greedy strategy shown in Algorithm 1 to obtain a 2-OPT solution.

# 3 ACTIVE LEARNING BASED MODEL TRAINING

## 3.1 OVERVIEW OF BALTO

BALTO is a pre-trained-model-based TPO approach that mainly consists of two parts, including AL-based model pre-training and pre-trained-model-based program optimization, shown in Figure 3. The model pre-training part is an active learning process that consists of four steps including *program sampling*, *model training*, *program selection*, and *performance evaluation*. Regarding step-1, BALTO samples programs randomly to form a large unlabeled dataset $D_u$ and a small labeled dataset $D_l$. Programs in labeled data set are measured on the hardware to obtain the labels. Then BALTO iterates step-2, step-3, and step-4 to train the model based on active learning. Regarding step-2, BALTO trains the model based on $D_l$. Regarding step-3, BALTO leverages the trained model and the labeled data set to select a batch of programs from $D_u$ for evaluation. Regarding step-4, the performance of programs are measured for updating $D_l$. Finally, the model training part outputs a trained model for later program optimization tasks once the total measurement budget is exhausted. The program optimization part takes the commonly used workflow as described in Section 2.1.

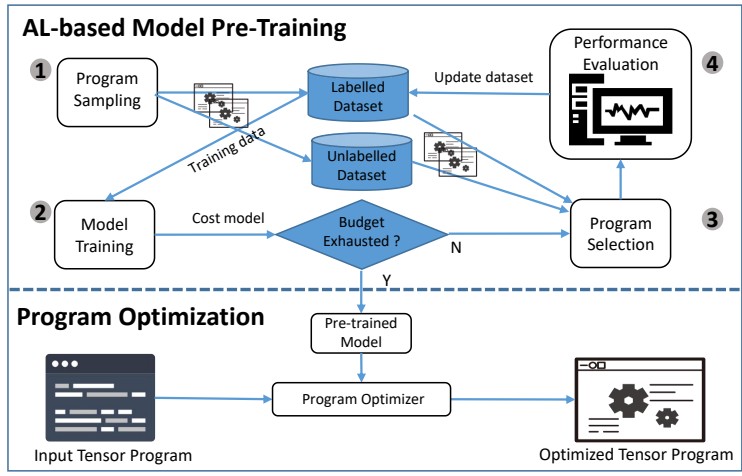

Figure 3: Overview of BALTO.

## 3.2 PROGRAM SAMPLING

BALTO generates unlabeled program data sets by randomly sampling transformed tensor programs from pre-defined optimization tasks. An optimization task corresponds to a machine learning operator (e.g., MatMul) with a specific shape, e.g., (1024, 1024, 1024). For a given task, programs can be randomly generated either by performing random transformations on program sketches(e.g., TenSet (Zheng et al., 2021) and Ansor (Zheng et al., 2020a)) or by sampling from probabilistic-language-defined stochastic search spaces (e.g., MetaSchedule (Shao et al., 2022)).

Random sampling can result in unbalanced distribution of the program performance. In such an unbalanced distribution, the number of low-performance programs accounts for a large part of the total samples, which makes the model easily fit to these low-performance programs instead of the

high-performance ones given a relatively small number of total samples. As a result, the prediction of the model for those high-performance programs falls into a very low accuracy, and thus cannot effectively guide the compiler to identify high-performance programs. Therefore, it is essential to balance the distribution of the samples by an appropriate program selection strategy that can increase the proportion of high-performance programs reasonably.

### 3.3 PROGRAM SELECTION

To balance the performance distribution of selected programs, BALTO leverages diversity-based selection to increase the diversity of selected programs' performance. We first propose an output-diversity-based selection scheme that maximizes the diversity of the predicted performance of selected programs based on core-set selection. However, the selection scheme suffers from a performance overestimation problem which make the distribution unbalanced even under the proposed selection scheme. Thus we further propose a biased-diversity-based selection to solve the problem.

**Output-diversity-based selection.** This selection scheme is based on the greedy selection scheme shown in Algorithm 1 with $d_{out}(x_i, x_j) = |f(x_i) - f(x_j)|$ where $f(x_i)$ is the predicted performance of program $x_i$. The main advantage of the distance function is that it can ensure a much more balanced distribution of the sampling. This is because the diversity of the predicted performance makes the predicted performance of the sampled programs more decentralized into the range [0,1], rather than centralized around 0, thus greatly reducing the proportion of low-performance programs.

**Performance overestimation.** For a given task, the maximal program throughput $T_{max}$ can be underestimated to a large extent under a small number of performance measurements. Correspondingly, the underestimated $T_{max}$ further incurs overestimated relative performances of most programs at the early stage of AL process. As a result, even with the diversity-based selection, such overestimation can still make the sampled programs imbalanced and mainly gathered at low-performance regions.

**Biased-diversity-based selection.** To solve the overestimation problem, we propose a biased selection process that can effectively select the high-performance programs in case of that the sampled programs are gathered at low-performance regions. Concretely, we assign a weight (i.e., $f(x_i)$) for program $x_i$ to encourage the algorithm to select programs with high scores on predicted performance, so as to accelerate the exploration of programs in high-performance regions. Thus the distance function becomes $d_{biased}(x_i, x_j) = f(x_i)d_{out}(x_i, x_j)$. Compared to the core-set selection that selects samples by solving the *k-center* problem, we formulate the biased selection as solving following *weighted k-center* problem:

$$\min_{S_t \subseteq D_u} \max_{x_i \in S_t} f(x_i) \min_{x_j \in D_l \cup S_t} d_{out}(x_i, x_j) \quad s.t. \quad |S_t| \leq B_t \tag{1}$$

Since the problem is NP-hard, we solve it via a greedy strategy as in Algorithm 2.

For each task, BALTO performs biased-diversity-based selection when the measured performances are gathered in the low-performance regions and performs output-diversity-based selection when the distributions are well balanced. Although the biased selection scheme can alleviate the imbalance of performance distribution caused by overestimation, it scarifies

---

**Algorithm 2** Biased-diversity-based Selection

**Require:** $D_l, D_u, B_t$
**Output:** $S_t$ is a set of samples to be labeled at iteration t
1: $S_t \leftarrow \phi$
2: **while** $|S_t| < B_t$ **do**
3:      $u = \arg\max_{x_i \in D_u} f(x_i) \min_{x_j \in D_l \cup S} |f(x_i) - f(x_j)|$
4:      $S_t \leftarrow S_t \cup \{u\}$
5: **return** $S_t$

---

the diversity of high-performance programs to a certain extent. To increase the diversity, BALTO leverages the KL-divergence $D_{KL}(P|Q) = -\mathbb{E}[\log(y_i)]$, where $y_i$ represents the relative performance of measured programs of the task, as a criterion for choosing the selection scheme. $P(z)$ with $p(1) = 1$ is a binomial distribution that describes a distribution of selected programs where all programs are high-performance. $Q(x_i)$ with $q(\{x_i \text{ is high-performance}\}) = y_i$ is a binomial distribution where a program with a relative performance of $y_i$ has a possibility of $y_i$ to be high-performance. $D_{KL}$ can effectively estimate how close the distribution of the currently sampled programs to $P$. As such, the lower $D_{KL}$ is, the closer the relative performance of the sampled programs is to 1 and the lower the diversity is. If $D_{KL}$ is greater than a constant threshold

(i.e., $C$), BALTO enables the biased-diversity-based selection scheme to select programs since the measured programs are gathered in the low-performance region. Otherwise, BALTO enables the output-diversity-based selection scheme since the overestimation problem is alleviated.

**Selection complexity.** BALTO trains the model for $T$ iterations. In each iteration, BALTO iterates over all of the $N$ tasks to select programs for measurement. For each task, the algorithm can select at most $B_t$ samples from $D_u$ that consists of $M$ programs of the given task. For a single selection of programs, the algorithm performs either the output-diversity-based selection or the biased-diversity-based selection according to $D_{KL}$. Both of the two selection schemes have a complexity of $O(M^2)$, thus the total selection complexity is $O(TNB_tM^2)$ or $O(BM^2)$ where $B$ is the total measurement budget. Take TenSet for example, the $B$ is 429,810, the $M$ is 4,000, and the total selection time is 10.6 minutes. Thus, the adopted selection strategy in BALTO does not bring additional overhead to the training process.

## 4 PERFORMANCE EVALUATION

We integrate BALTO into TenSet to evaluate the performance comprehensively. The experimental methodology is five-fold. First, we show that BALTO achieves comparable accuracy while with much less hardware measurements on 6 platforms. Second, we demonstrate that the models trained with much less performance measurements have comparable optimization ability on a real-world hardware platform. Third, we demonstrate that BALTO is a generic optimization approach by integrating it to MetaScheduler. Fourth, we perform ablation study to verify the effectiveness of the core components of BALTO. Fifth, we visualize the performance distribution of our proposed sampling strategy to illustrate that the proposed strategy helps balancing the distribution of programs.

**Baselines.** To demonstrate the effectiveness of BALTO, we compare BALTO with five baselines including: 1) TenSet trains the model by measuring all the randomly sampled programs; 2) GSx (Yu & Kim, 2010) selects programs by maximizing the feature diversity greedily; 3) GSy (Wu et al., 2019) selects programs by maximizing the label diversity greedily; 4) iGS (Wu et al., 2019) selects programs by maximizing the label and feature diversity greedily; 5) ALT (Zeng et al., 2020) selects programs by the uncertainty of the model prediction. The proposed output-diversity-based selection is represented as Ours1 and the biased-diversity-based selection is presented as Ours2.

### 4.1 COMPARISON OF BALTO AND TENSET FOR MODEL PRECISION

**Dataset.** We evaluate BALTO's effectiveness on the dataset provided by TenSet. The dataset consists of program performance measurement records from 6 different hardware platforms. Each platform includes a total number of 8,596,208 tensor program measurement records that are sampled from 2307 different types of tasks. We use 10% of the records as the test dataset and the remaining 90% records as the train dataset for the baselines. We further select at most 5% of the training dataset for training BALTO and other active learning approaches (i.e., GSx, GSy, iGS, and ALT). We report the Top-1 score on the test dataset based on 5 independent experiments.

**Results.** We compare BALTO with baselines on two ML models including XGBoost and MLP. All the models adopt the same hyperparamenters with TenSet. As shown in Table 1, BALTO outperforms all other active learning baselines and achieves the same or even higher accuracy compared with TenSet. Compared to TenSet, BALTO reduces the hardware measurements by $20\times$. Take T4 for example, BALTO achieves an accuracy improvement of 4.1% on XGBoost model and achieves the same accuracy on MLP model respectively. The reason that TenSet achieves a relatively lower accuracy is that TenSet trains on a randomly sampled program dataset where the ratios of low-performance programs and high-performance programs are extremely imbalanced. Differently, such imbalance can be greatly alleviated by the proposed biased-diversity-based active learning of BALTO, thus delivering much lower training cost but similar or even higher accuracy.

### 4.2 OPTIMIZATION PERFORMANCE ON A REAL-WORLD PLATFORM

To verify that cost models trained on much fewer programs can still effectively guide the TPO process, we evaluate BALTO on NVIDIA T4 and report the optimized execution time of 5 commonly used neural networks of BALTO and the baselines.

Table 1: Comparison between BALTO and TenSet on 6 hardware platforms with 2 learning models. GPU-1 and GPU-2 represents NVIDIA T4 and NVIDIA K80, respectively. CPU-1, CPU-2, CPU-3, and CPU-4 represents Intel Platinum 8272CL, Intel E5-2673 v4, AMD EPYC 7452, and ARM Graviton2, respectively.

| XGBoost | GPU-1 | GPU-2 | CPU-1 | CPU-2 | CPU-3 | CPU-4 |
|---|---|---|---|---|---|---|
| TenSet | $84.7 \pm 0.4$ | $84.9 \pm 0.1$ | $82.7 \pm 0$ | $83.0 \pm 0$ | $83.9 \pm 0$ | $79.2 \pm 0$ |
| GSx | $81 \pm 0$ | $80.9 \pm 0$ | $78.1 \pm 0.27$ | $78 \pm 0$ | $79.6 \pm 1.01$ | $75.6 \pm 0$ |
| GSy | $81.3 \pm 0$ | $79.8 \pm 0$ | $78.8 \pm 0$ | $80 \pm 0$ | $81.4 \pm 0$ | $76.5 \pm 0$ |
| iGS | $82.3 \pm 0$ | $80.9 \pm 0$ | $79 \pm 0$ | $80.1 \pm 0.2$ | $81.4 \pm 0.3$ | $76.8 \pm 0.7$ |
| ALT | $85.3 \pm 0$ | $85.2 \pm 0$ | $81.5 \pm 0$ | $83.7 \pm 0$ | $84.6 \pm 0$ | $78.8 \pm 0$ |
| **Ours1** | $87.3 \pm 0.5$ | $87.6 \pm 0$ | $86.2 \pm 0$ | $85.6 \pm 0$ | $87.3 \pm 0.1$ | $80.6 \pm 0$ |
| **Ours2** | $\mathbf{88.8} \pm 0.2$ | $\mathbf{88.9} \pm 0.1$ | $\mathbf{87.3} \pm 0.3$ | $\mathbf{86.4} \pm 0.5$ | $87.9 \pm 0.2$ | $\mathbf{81.6} \pm 0.1$ |
| **MLP** | **GPU-1** | **GPU-2** | **CPU-1** | **CPU-2** | **CPU-3** | **CPU-4** |
| TenSet | $90.5 \pm 0.3$ | $90.2 \pm 0.5$ | $86.5 \pm 0.8$ | $86.1 \pm 0.4$ | $88.1 \pm 0.5$ | $80.8 \pm 0.8$ |
| GSx | $83.5 \pm 1.6$ | $83.3 \pm 0.9$ | $76.2 \pm 0.2$ | $74.8 \pm 0.3$ | $77.6 \pm 1.7$ | $69.5 \pm 0.7$ |
| GSy | $82.9 \pm 1.9$ | $82.2 \pm 0.3$ | $75.1 \pm 1.1$ | $72.9 \pm 1.6$ | $74.7 \pm 0.4$ | $68.9 \pm 1.5$ |
| iGS | $82.3 \pm 0.4$ | $81.4 \pm 0.2$ | $74.1 \pm 2.2$ | $73.9 \pm 2$ | $75.3 \pm 0.9$ | $69.2 \pm 2.4$ |
| ALT | $86 \pm 0.4$ | $84.3 \pm 0.2$ | $77.3 \pm 1.2$ | $79.2 \pm 1.2$ | $75.3 \pm 0.9$ | $69.2 \pm 2.4$ |
| **Ours1** | $89.6 \pm 0.9$ | $89.8 \pm 0.7$ | $86 \pm 0.4$ | $86.4 \pm 0.8$ | $88.1 \pm 0.6$ | $79 \pm 1.1$ |
| **Ours2** | $90.4 \pm 0.6$ | $90.3 \pm 0.6$ | $86.9 \pm 1.0$ | $\mathbf{87.2} \pm 1.1$ | $\mathbf{89.1} \pm 1.3$ | $81.1 \pm 1.1$ |

**Benchmarks.** We evaluate the optimization results on 5 commonly used neural networks including ResNet-50 (He et al., 2016), MobileNet-v2 (Sandler et al., 2018), ResNext-50 (Xie et al., 2017), BERT-tiny and BERT-base (Devlin et al., 2018). For the three CNN models, we set the batch size to 1 and the input shape to $224 \times 224$. For BERT models, we set the sequence length to be equal to 128. We assign at most 1,000 trials of hardware measurements for each network and report the average execution time based on 5 independent experiments of program optimization.

Table 2: Evaluations of BALTO and TenSet on 5 commonly used neural networks.

| XGBoost | ResNet-50 | MobileNet-v2 | ResNext-50 | BERT-tiny | BERT-base |
|---|---|---|---|---|---|
| TenSet | $3.89 \pm 0.4$ ms | $0.60 \pm 0.04$ ms | $3.41 \pm 0.31$ ms | $3.06 \pm 0.30$ ms | $10.8 \pm 0.2$ ms |
| GSx | $3.92 \pm 0.1$ ms | $0.62 \pm 0.08$ ms | $3.62 \pm 0.09$ ms | $3.11 \pm 0.05$ ms | $12.2 \pm 0.7$ ms |
| GSy | $4.39 \pm 0.4$ ms | $0.63 \pm 0.05$ ms | $3.65 \pm 0.56$ ms | $3.03 \pm 1.10$ ms | $11.1 \pm 1.1$ ms |
| iGS | $3.81 \pm 0.1$ ms | $0.65 \pm 0.03$ ms | $3.71 \pm 0.73$ ms | $3.26 \pm 0.21$ ms | $11.0 \pm 1.0$ ms |
| ALT | $3.62 \pm 0.1$ ms | $0.59 \pm 0.02$ ms | $3.30 \pm 0.04$ ms | $2.81 \pm 0.13$ ms | $11.1 \pm 1.6$ ms |
| **Ours1** | $3.68 \pm 0.5$ ms | $0.59 \pm 0.06$ ms | $3.27 \pm 0.1$ ms | $2.84 \pm 0.09$ ms | $10.9 \pm 0.3$ ms |
| **Ours2** | $\mathbf{3.30} \pm 0.2$ ms | $0.58 \pm 0.04$ ms | $\mathbf{3.23} \pm 0.17$ ms | $\mathbf{2.76} \pm 0.05$ ms | $10.8 \pm 0.4$ ms |
| **MLP** | **ResNet-50** | **MobileNet-v2** | **ResNext-50** | **BERT-tiny** | **BERT-base** |
| TenSet | $3.28 \pm 0.15$ ms | $0.57 \pm 0.03$ ms | $3.26 \pm 0.32$ ms | $2.80 \pm 0.09$ms | $10.0 \pm 0.4$ ms |
| GSx | $4.13 \pm 0.17$ ms | $0.64 \pm 0.08$ ms | $3.50 \pm 0.32$ ms | $3.16 \pm 0.13$ ms | $11.5 \pm 0.8$ ms |
| GSy | $3.79 \pm 0.58$ ms | $0.62 \pm 0.07$ ms | $3.40 \pm 0.24$ ms | $2.92 \pm 0.15$ ms | $11.5 \pm 0.5$ ms |
| iGS | $3.58 \pm 0.09$ ms | $0.62 \pm 0.06$ ms | $3.43 \pm 0.13$ ms | $2.91 \pm 0.01$ ms | $11.3 \pm 1.1$ ms |
| ALT | $4.04 \pm 0.25$ ms | $0.75 \pm 0.19$ ms | $3.30 \pm 0.16$ ms | $3.06 \pm 0.16$ ms | $11.6 \pm 0.9$ ms |
| **Ours1** | $3.43 \pm 0.42$ ms | $0.60 \pm 0.02$ ms | $3.30 \pm 0.33$ ms | $2.79 \pm 0.07$ ms | $10.8 \pm 0.5$ ms |
| **Ours2** | $3.33 \pm 0.17$ ms | $0.58 \pm 0.00$ ms | $3.14 \pm 0.17$ ms | $2.71 \pm 0.13$ ms | $9.9 \pm 0.4$ ms |

**Results.** As shown in Table 2, BALTO achieves the same or better optimization performance compared to the baselines. Specially, when using the XGBoost cost model, BALTO achieves an average of $1.07\times$ performance improvement over TenSet. The improvement comes from a more accurate performance model that helps the compiler to identify the high-performance programs efficiently. When using the MLP cost model, the performance of BALTO is the same as that of TenSet since the two approaches' models have the same accuracy on T4. The experiments' results show that models trained on much fewer measured programs can still guide the optimization efficiently. Thus, we do not need to measure as much of programs as existing optimization frameworks do.

## 4.3 COMPATIBILITY OF BALTO

BALTO is a generic pre-trained-model-based tensor program optimization approach that is compatible with other TPO frameworks. To demonstrate this, we integrate BALTO into MetaScheduler (Shao et al., 2022) to perform auto-tensorization using TensorIR (Feng et al., 2022). Take the GEMM operator for example, we randomly sample 800,000 unmeasured programs at first. To train the model, we then use BALTO to measure at most 5% of the total programs on an Intel Xeon Gold 6226R CPU. To verify the optimization ability of the trained model, we generate 80 tasks with different problem sizes (i.e., $N = 512 + 64 * i, i \in$

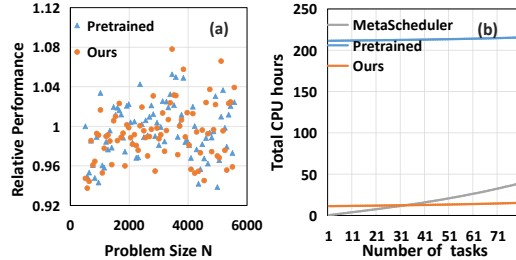

Figure 4: Comparison of optimized performance and tuning time. (a) Optimized performances of GEMM operators with shape $(N, N, N)$. (b) Total tuning time.

$0, 1, .., 79$). We experiment on two baselines including MetaScheduler and pre-trained-model-based MetaScheduler (i.e., the *Pretrained*). The MetaScheduler guides the search with a model trained from scratch and the *Pretrained* trains the model by measuring all the 800,000 programs. For a given task, we limit the total trials of MetaScheduler to 1000 and the *Pretrained* as well as BALTO to 100. Regarding the optimization ability, we compare the relative performance which is normalized to MetaScheduler. As shown in Figure 4(a), the relative performance of BALTO to MetaScheduler is between $0.93\times$ and $1.08\times$ with $0.992\times$ on average, which demonstrates that BALTO achieves similar optimization ability. Regarding the tuning time, BALTO reduces the pre-training overhead by $20\times$ and the single task optimization time by $10\times$. As shown in Figure 4(b), the total tuning time of MetaScheduler grows rapidly with the number of optimization tasks while our tuning time grows much slower.

## 4.4 ABLATION STUDY

**Effectiveness of biased selection.** As shown in Figure 5, biased selection with a proper $C$ (e.g., $C = 1$) can outperform the output-diversity-based approach and the random selection approach. At the early stage of the training, the accuracy score of the biased selection approach grows much faster than that of the output-diversity-based approach, since the bias for high-performance programs effectively reduces the distribution imbalance caused by the overestimation problem. Meanwhile, after the early stage of the training, the estimation of the relative performance becomes more accurate as well as the distribution becomes more balanced. Therefore, the biased selection switches to the output-diversity-based selection and finally achieves a higher precision score under the same budget.

**Optimization ability on sub-graphs.** As shown in Figure 6 and Figure 7, BALTO achieves comparable optimization ability with TenSet on two representative sub-graphs including batched matmul and 2D convolution. Both BALTO and TenSet converge faster than Ansor since the pre-trained models are accurate enough to guide the search without massive hardware measurements. For sub-graphs (i.e., batched matmul) with large search spaces, Ansor requires more trials (i.e., greater than 1000) to converge since it searches with an inaccurate model that is trained from scratch. Meanwhile, the pre-trained-model-based approaches can search such spaces more quickly and efficiently under the guidance of an accurate model.

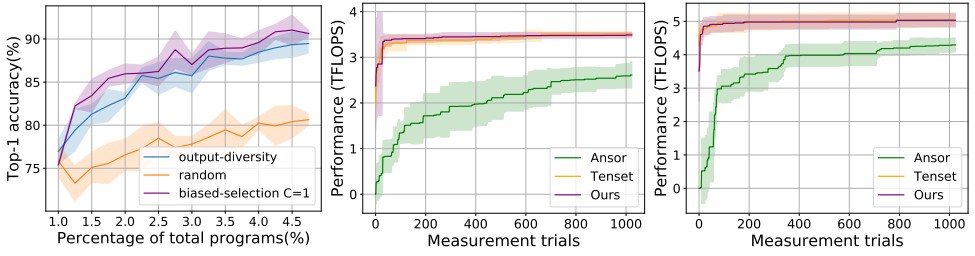

Figure 5: Comparison of different selection schemes.

Figure 6: Batched matmul.

Figure 7: 2D convolution.

### 4.5 VISUALIZATION

**Visualization of the performance distributions.** Figure 8 is a histogram of program performance distributions at two different time steps (e.g., the first time step and the last time step). The y-axis represents the percentage of programs and the x-axis represents the relative performance. As shown in Figure 8, the program distribution of the first time step is extremely unbalanced since the programs are sampled randomly. After 15 steps of program sampling based on active learning, the program distribution of the last time step becomes more balanced, which clearly demonstrates the effectiveness of our biased-diversity-based AL approach.

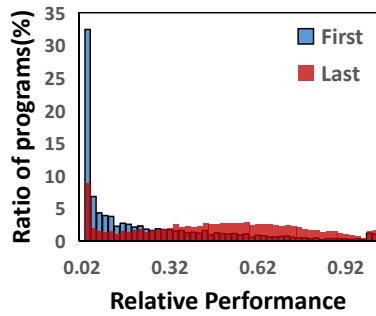

Figure 8: Performance distributions.

## 5 RELATED WORK

**Tensor optimization frameworks.** Commonly used TPO frameworks include online-trained-model-guided program optimization (Chen et al., 2018; Zheng et al., 2020a; 2022; Ahn et al., 2020; Zeng et al., 2020), and predefined-model-guided program optimization (Zhu et al., 2022; Dave et al., 2019; Adams et al., 2019; Haj-Ali et al., 2020; Anderson et al., 2020; Zheng et al., 2021). Predefined-model-guided program optimization can optimize the given tensor programs in a short time and its predefined model can be either designed manually by domain experts (Zhu et al., 2022; Dave et al., 2019) or pre-trained using randomly generated programs (Adams et al., 2019; Haj-Ali et al., 2020; Anderson et al., 2020; Zheng et al., 2021). The manually designed model are usually designed for specific hardware platforms and thus hard to adapt to new platforms. The pre-trained-model based approaches can effectively adapt to the new platforms but require a large number of hardware measurements of randomly sampled programs. The online-trained-model-guided approaches relies training the model with data collected online. Thus this kind of optimization approaches is time consuming for each input optimization task. Although BALTO is a fast tensor optimization approach designed for the pre-trained-model-based approaches, the optimization schemes in BALTO can also be applied to the online-trained-model-guided approaches, which is our future work.

**Active learning.** AL is widely used for reducing the expensive annotation cost for model training (Aghdam et al., 2019; Joshi et al., 2009; Culotta & McCallum, 2005). The acquisition function can reduce the redundancy in sampling by only anotating informative or representative samples and thus is the core to AL. Typically, the most frequently used acquisition strategies can be divided into uncertainty-based approaches (Yoo & Kweon, 2019; Roth & Small, 2006) and diversity-based approaches (Sener & Savarese, 2018; Hasan & Roy-Chowdhury, 2015). Uncertainty-based approaches tend to query samples that are most uncertain (Lewis & Catlett, 1994). MC Dropout (Gal & Ghahramani, 2016)adopts random dropout for obtaining CNN's prediction uncertainty. (Beluch et al., 2018) uses an ensemble of classifiers for obtaining the uncertainty. Diversity-based acquisition emphasizes on the diversity of selected samples. GSx (Yu & Kim, 2010) maximizes the diversity in the feature space, while iGS (Wu et al., 2019) maximizes the diversity in both feature space and label space. Our active learning approach is based on the Core-Set framework and explores the diversity of predicted programs' performance.

## 6 CONCLUSION

In this paper, we propose BALTO, a fast TPO approach with biased-diversity-based active learning to reduce the training cost of the pre-trained model while with the same or higher prediction accuracy. The key insight is that there exists a heavy redundancy of low-performance programs when sampling randomly with existing approaches. To remove such redundancy and for a much lower training cost, active learning is introduced in BALTO. Moreover, we further propose a biased-diversity-based scheme specially designed for BALTO to address the overestimation problem. The empirical results demonstrate that BALTO consistently shows superior training performance (i.e., $20\times$ reduction in required hardware measurements.) on a wide range of environment configurations (i.e., 6 hardware platforms and 2 learning models) and even better optimization performance on T4.

## ACKNOWLEDGMENTS

We would like to thank the reviewers for their valuable suggestions. This work is partially supported by the National Key R&D Program of China (under Grant 2021ZD0110102), the NSF of China (under Grants U22A2028, 61925208, 62102399, 62002338, 62222214, U19B2019), CAS Project for Young Scientists in Basic Research (YSBR-029), Youth Innovation Promotion Association CAS and Xplore Prize.

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

## A  APPENDIX

You may include other additional sections here.

