# OpenReview forum: "BALTO: fast tensor program optimization with diversity-based active learning"
_ICLR.cc/2023/Conference — ICLR 2023 poster_

### Official Review · Reviewer_nyuX · 2022-10-14

**Confidence:** 4
**Correctness:** 3
**Technical Novelty And Significance:** 3
**Empirical Novelty And Significance:** 3
**Recommendation:** 6

**Clarity, Quality, Novelty And Reproducibility:**

- Clarity & qualty. The paper is clearly written and easy to follow. With rich insights presented, the reviewer believes the paper provides a promising direction for future exploration. A quick side note: given it’s written as an insightful analysis paper, it would be even more clear to avoid unnecessary formula, for example, Equation (1)(2), and present the insights in a plain text that readers don’t have to think twice.
- Novelty. The paper tackles a long-existing problem in a novel way, by creatively applying and improving active learning according to the insights and observation in tensor program optimization.
- Reproducibility. Given the limited time budget, the reviewer does not have the bandwidth to evaluate the artifact or code.


**Strength And Weaknesses:**

Strength
* As a practitioner of deep learning compilation, I am enthusiastic about this paper because of the accurate analysis presented in this paper. While active learning does not necessarily have to be “the only” approach, as the solution proposed the paper, it apparently does solve the problem revealed by the analysis, and most importantly, the insights presented in the paper provides a promising direction for follow-up works to further explore.
* The output-diversity-based program selection, plus biased-diversity-based active learning, are quite novel and invented with deep understanding of the characteristics of tensor program optimization. For example, algorithm in Section 3.3 comes from very careful analysis that the estimated maximal program throughput could be underestimated, and thus it’s used to avoid overestimation of relative performance.
* Experiments are designed carefully to align with numbers in the baseline, i.e. TenSet, and much better performance is showcased in those very well aligned experiments.

Weakness
* While experimented in several standard representative models, given the proposed solution is algorithmcally generic, it would be desirable to demonstrate that it could be further extended to more complicated scenarios, for example, automatic tensorization in TensorIR [1], auto tuning with sparse computation [2]. The reviewer doesn’t believe it’s any major weakness, but is curious if it could be generic enough to generalize to broader usecases.
* More ablation study is desirable to present with more in-depth the effectiveness of such exploration algorithm in different subgraphs, for example, the difference between batched matmul in BERT and winograd convolution in ResNet; On the other hand, the characteristics of models selected in Table 2 might overlap, for example, between BERT-base and BERT-tiny - is there any reason to have both in the table?
* The relationship of some lines of work is worth more discussion. For example, is this algorithm compatible with the programming model proposed in MetaSchedule [3]? What samples does the resulting models favor, compared with human prior-based approaches like Roller [4]?

[1] Feng, Siyuan, Bohan Hou, Hongyi Jin, Wuwei Lin, Junru Shao, Ruihang Lai, Zihao Ye et al. "TensorIR: An Abstraction for Automatic Tensorized Program Optimization." arXiv preprint arXiv:2207.04296 (2022).

[2] Ye, Zihao, Ruihang Lai, Junru Shao, Tianqi Chen, and Luis Ceze. "SparseTIR: Composable Abstractions for Sparse Compilation in Deep Learning." arXiv preprint arXiv:2207.04606 (2022).

[3] Shao, Junru, Xiyou Zhou, Siyuan Feng, Bohan Hou, Ruihang Lai, Hongyi Jin, Wuwei Lin, Masahiro Masuda, Cody Hao Yu, and Tianqi Chen. "Tensor Program Optimization with Probabilistic Programs." arXiv preprint arXiv:2205.13603 (2022).

[4] Zhu, Hongyu, Ruofan Wu, Yijia Diao, Shanbin Ke, Haoyu Li, Chen Zhang, Jilong Xue et al. "{ROLLER}: Fast and Efficient Tensor Compilation for Deep Learning." In 16th USENIX Symposium on Operating Systems Design and Implementation (OSDI 22), pp. 233-248. 2022.

**Summary Of The Paper:**

This paper presents a novel active learning-based approach that accurately addresses an long-existing pain point in deep learning compilation, namely tuning speed. Its contribution could be summarized as:
* Presented a careful investigation into the root cause of the unnecessarily long tuning speed, which is lack of diversity in sampling that leads to failure of comprehensive exploration of the entire search space, while large chunk of exploratory trials are wasted in apparently low-quality candidates;
* A bias-diversity-based active learning-based approach is proposed to mitigate the problem revealed by the investigation. First, it applies the existing methodology of diversity-based approach by carefully controlling the sample generation to be aware of the similarity between samples; Then, a bias selection scheme is used to alleviate the imbalanced distribution that diversity-based approach could lead to (in terms of performance);
* Experiments on popular models are shown to demonstrate the superior performance of the proposed solution.


**Summary Of The Review:**

In summary, the reviewer believes that the paper makes significant contribution in terms of insightful investigation of the weakness of existing sampling process, namely over-sampling of weak candidates (imbalanceness) and underestimation of the peak performance (estimated maximal throughput). The concrete active learning-based algorithm, as proposed in this paper, properly solves the issue being found. This work is particularly valuable in terms of its insights and future direction it leads to.

The reviewer is willing to adjust the score accordingly if more diverse setting of experiments and more detailed comparison between this work and previous lines of research is present.

---

> ### Author Response · Authors · 2022-11-16
> **Response to nyuX**
>
> We thank reviewer-nyuX for your appreciation and constructive suggestions. We provide a detailed response below.
>
> **Q1: While experimented in several standard representative models, given the proposed solution is algorithmically generic, it would be desirable to demonstrate that it could be further extended to more complicated scenarios, for example, automatic tensorization in TensorIR, auto tuning with sparse computation.**
>
>
> A1: BALTO is a generic solution for TPO based on pre-trained models. To demonstrate this, we integrate it into MetaScheduler which leverages TensorIR to perform auto-tensorization. The new experimental results are shown in Section 4.3. We randomly sampled 200 optimization tasks (i.e., 800,000 programs in total) for a GEMM operator and trained  BALTO on these tasks. Then we evaluate the optimization ability on 80 new tasks that are not in the training dataset. The relative performance of BALTO to MetaScheduler is between $0.93\times$ and $1.08\times$ with $0.992\times$ on average, which demonstrates that BALTO achieves similar optimization ability. Regarding the tuning time, BALTO reduces the pre-training overhead by $20\times$ (compared to pre-trained-model-based MetaScheduer )and the single task optimization time by $10\times$ (compared to MetaScheduer), which demonstrates that BALTO can greatly improve the tuning speed.
>
> **Q2: More ablation study is desirable to present with more in-depth the effectiveness of such exploration algorithm in different subgraphs; On the other hand, the characteristics of models selected in Table 2 might overlap, for example, between BERT-base and BERT-tiny - is there any reason to have both in the table?**
>
> A2: We have added the ablation study of two sub-graphs including Batched matmul and 2D convolution in Section 4.4. We compared BALTO with TenSet and Ansor (i.e., TenSet with a model trained from scratch). The experimental results (i.e., Figure 6 and Figure 8) show that BALTO has a similar optimization ability with TenSet and outperforms Ansor in both the convergence speed and the optimized performance on these two sub-graphs. Regarding the network baselines, we follow the same configuration of TenSet for a fair comparison.
>
> **Q3: The relationship of some lines of work is worth more discussion. For example, is this algorithm compatible with the programming model proposed in MetaScheduler? What samples do the resulting models favor, compared with human prior-based approaches like Roller?**
>
> A3: BALTO is a generic approach for pre-trained-model-based TPO and is compatible with the programming model proposed in MetaScheduler. To demonstrate this, we integrate BALTO with MetaScheduler which leverages TensorIR to perform auto-tensorization. The experimental results well demonstrate the effectiveness of BALTO. Roller guides the search by an expert-designed model which evaluates programs  according to their tile sizes. BALTO guides the search by a data-driven model which evaluates programs according to more program features (e.g., unroll length, vector length, etc). Compared to Roller, BALTO's cost model requires no human efforts and thus is more flexible to support new platforms.
>
> **Q4: A quick side note: given it’s written as an insightful analysis paper, it would be even more clear to avoid unnecessary formulas, for example, Equation (1)(2), and present the insights in a plain text that readers don’t have to think twice.**
>
> A4: We have revised the paper to avoid unnecessary formulas. Concretely, we have moved the introduction of core-set selection to the background in Section 2.3. We have removed the original Equation(2) to make it clearer. We have also added a new subsection for introducing the insight of performance overestimation. For a given task, the maximal program throughput $T_{max}$ can be underestimated to a large extent under a small number of performance measurements. Correspondingly, the underestimated $T_{max}$ further incurs in overestimated relative performances of most programs at the early stage of AL process. As a result, even with the diversity-based selection, such overestimation can still make the sampled programs imbalanced and mainly gathered at low-performance regions.
>
> We hope that our response has mostly addressed the reviewer’s concerns. We are happy to continue a discussion to address any other questions the reviewer may have.

---

### Official Review · Reviewer_7kGb · 2022-10-24

**Confidence:** 4
**Correctness:** 2
**Technical Novelty And Significance:** 2
**Empirical Novelty And Significance:** Not applicable
**Recommendation:** 5

**Clarity, Quality, Novelty And Reproducibility:**

The writing in the paper is not always clear. There are many places where words need to be added in to make the sentences clear.

The quality of analysis as outlined above needs to be improved.

The novelty of the work does not seem that great given the fact that works exist which use Active Learning and are not discussed here. Nor is the work compared to other optimisation techniques used.

**Strength And Weaknesses:**

The introduction to the paper fails to set the context for the work making it difficult to follow for all but those who know the area well. By contrast the second section gives a very low-brow overview of the area. A shortened version of this should be merged into the introduction thus increasing space for other material.

The use of active learning for TPO is not a new idea. A simple google search reveals "ALT: Optimizing Tensor Compilation in Deep Learning Compilers with Active Learning". It is very surprising that the authors do not compare themselves with this work. It does not appear in the related work nor is it used to compare results with. To clam TenSet is state-of-the-art is rather misleading. A full comparison with other Active Learning approaches should be performed in order to justify if this work is better than these other works. ATL was not the only AL paper out there.

For the results there was no indication of whether the results were from one run or averaged over many. There has been a lot of criticism in AI recently for authors cherry-picking results and the authors here should show that their results are true irrespective of runs or random seeds.

**Summary Of The Paper:**

The paper identifies that with Tensor Programming Optimisation in general the dataset collected is significantly imbalanced. Using this they develop an active learning approach which favours the selection of new samples as those which are hoped to be in areas with less prior knowledge. The authors show that this allows them to gain the same performance with far fewer samples.

**Summary Of The Review:**

The paper seems a marginal improvement in the area - though without comparison to the real state-of-the-art work it is difficult to judge if there is a benefit to this approach or not. The work could benefit from more analytical analysis of the results. Many of the equations are presented without any justification or motivation as to why they may be beneficial.

---

> ### Author Response · Authors · 2022-11-16
> **Response to 7kGb : PART 2**
>
> **Q3: To claim TenSet is state-of-the-art is rather misleading. A full comparison with other Active Learning approaches should be performed in order to justify if this work is better than these other works.**
>
>  A3: To our best knowledge, we are the first work to introduce active learning for pre-trained-model-based TPO which is the state-of-the-art approach for reducing the optimization overhead of TPO. For TPO based on pre-trained model, Tenset is one of the state-of-the-art approaches which reduce the optimization time by up to $10\times$. ALT performs active learning for TPO with a model trained from scratch and achieves up to $2.49\times$ speedup in the optimization time. Compared to ALT, TenSet and BALTO solve a different problem where pre-trained models are used to guide the search.
> Although ALT does not support TPO based on pre-trained model, we integrate its active learning method into our framework as a baseline.
>  As shown in Table 1 and Table 2, BALTO outperforms ALT on all the experimental configurations.
>  Also, we have compared BALTO with other active learning approaches designed for regression problems. The effectiveness of BALTO comes from the ability to balance the performance distribution of selected programs.
>
> We have added four active learning baselines, which are designed for regression problems, as follows:
> 1) GSx [1] selects programs by maximizing the feature diversity  greedily;
> 2) GSy [2] selects programs by maximizing the label diversity greedily;
> 3) iGS [2] selects programs by maximizing the label and feature diversity  greedily;
> 4) ALT [3] selects programs by the uncertainty of the model prediction.
>
>  The experimental results of other active learning approaches on model accuracy are summarized in following table (the higher the better):
>  |XGBoost|GPU-1|GPU-2|CPU-1|CPU-2|CPU-3|CPU-4|
> |-------|----|----|----|----|----|----|
> |GSx|$81\pm0$|$80.9\pm0$|$78.1\pm0.27$|$78\pm0$|$79.6\pm1.01$|$75.6\pm0$|
> |GSy|$81.3\pm0$|$79.8\pm0$|$78.8\pm0$|$80\pm0$|$81.4\pm0$|$76.5\pm0$|
> |iGS|$82.3\pm0$|$80.9\pm0$|$79\pm0$|$80.1\pm0.2$|$81.4\pm0.3$|$76.8\pm0.7$|
> |ALT|$85.3\pm0$|$85.2\pm0$|$81.5\pm0$|$83.7\pm0$|$84.6\pm0$|$78.8\pm0$|
> |Ours|88.8$\pm0.2$|88.9$\pm0.1$|87.3$\pm0.3$|86.4$\pm0.5$|87.9$\pm0.2$|81.6$\pm0.1$|
> |**MLP**|**GPU-1**|**GPU-2**|**CPU-1**|**CPU-2**|**CPU-3**|**CPU-4**|
> |GSx|$83.5\pm1.6$|$83.3\pm0.9$|$76.2\pm0.2$|$74.8\pm0.3$|$77.6\pm1.7$|$69.5\pm0.7$|
> |GSy|$82.9\pm1.9$|$82.2\pm0.3$|$75.1\pm1.1$|$72.9\pm1.6$|$74.7\pm0.4$|$68.9\pm1.5$|
> |iGS|$82.3\pm0.4$|$81.4\pm0.2$|$74.1\pm2.2$|$73.9\pm2$|$75.3\pm0.9$|$69.2\pm2.4$|
> |ALT|$86\pm0.4$|$84.3\pm0.2$|$77.3\pm1.2$|$79.2\pm1.2$|$75.3\pm0.9$|$69.2\pm2.4$|
> |Ours|$90.4\pm0.6$|$90.3\pm0.6$|$86.9\pm1.0$|87.2$\pm1.1$|89.1$\pm1.3$|$81.1\pm1.1$|
>
> [1] Hwanjo Yu and Sungchul Kim. Passive sampling for regression. 2010 IEEE International Conference on Data Mining, pp. 1151–1156, 2010.
>
> [2] Dongrui Wu, Chin-Teng Lin, and Jian Huang. Active learning for regression using greedy sampling.
>
> [3] Xi Zeng, Tian Zhi, Zidong Du, Qi Guo, Ninghui Sun, and Yunji Chen. Alt: Optimizing tensor compilation in deep learning compilers with active learning. In 2020 IEEE 38th International Conference on Computer Design (ICCD), pp. 623–630. IEEE, 2020.
>
> **Q4: For the results, there was no indication of whether the results were from one run or averaged over many.**
>
> A4: We understand your concern and have modified Table 1 and Table 2 correspondingly. Each result is averaged on five independent experiments with a 95\% confidence interval.
>
> We hope that our response has mostly addressed the reviewer’s concerns. We are happy to continue a discussion to address any other questions the reviewer may have.

---

> ### Author Response · Authors · 2022-11-16
> **Response to 7kGb : PART 1**
>
> We thank reviewer-7kGb for the valuable suggestions. We provide a detailed response below.
>
> **Q1: The introduction to the paper fails to set the context for the work making it difficult to follow for all but those who know the area well.**
>
>  A1: Our work aims at reducing the prohibitively expensive training overhead of pre-trained-model-based tensor program optimization. We have revised the first paragraph of the introduction to make the paper easier to follow. Tensor program optimization (TPO) can effectively reduce the computing time of neural networks by searching for high-performance programs in a designed search space. In TPO, neural networks are first represented as a set of tensor programs that describe the computation of multi-dimensional data arrays. Then performances of these tensor programs are measured on a target hardware platform.
> Such hardware measurements are time-consuming and thus optimizing a given network can cost several days or even weeks, which greatly hinders the wide application of TPO. To accelerate TPO, pre-trained machine learning (ML) models are introduced to replace a substantial part of the hardware measurements with performance predictions of a pre-trained model. For example, as the state-of-the-art approach, TenSet can significantly reduce the optimization time by up to $10\times$ through training on a large-scale and well-established dataset.
>
> **Q2: The work could benefit from more analytical analysis of the results. Many of the equations are presented without any justification or motivation as to why they may be beneficial.**
>
> A2: Thanks for your suggestion, we have added more analysis of the equations.
> Regarding the output-diversity-based selection, the reason was originally described in Section 3.3. The main advantage of the distance function (i.e.,  $d_{out}(x_{i}, x_{j})= |f(x_{i}) - f(x_{j})|$) is that it can ensure a much more balanced distribution of the sampling. This is because the diversity of the predicted performance makes the predicted performance of the sampled programs more decentralized into the range [0,1], rather than centralized around 0, thus greatly reducing the proportion of low-performance programs.
> Regarding the biased-diversity-based selection, we describe our motivation as follows. We assign a weight (i.e., $f(x_i)$) for the program $x_i$ to encourage the algorithm to select programs with high scores on predicted performance, so as to accelerate the exploration of programs in high-performance regions. Thus the distance function becomes $d_{biased}(x_{i}, x_{j})= f(x_{i})d_{out}(x_i, x_j)$. Compared to the core-set selection that selects samples by solving the k-center problem, we formulate the biased selection as solving a weighted k-center problem (i.e., $\min \limits_{S_{t} \subseteq D_u} \max \limits_{x_{i} \in S_{t}} f(x_{i})  \min \limits_{x_{j} \in D_l \cup S_{t}} d_{out}(x_{i}, x_{j})\quad s.t. \quad |S_{t}| \le B_{t}$).

---

### Official Review · Reviewer_3pkd · 2022-10-25

**Confidence:** 4
**Correctness:** 3
**Technical Novelty And Significance:** 3
**Empirical Novelty And Significance:** 4
**Recommendation:** 8

**Clarity, Quality, Novelty And Reproducibility:**

Novelty: The paper explores a novel avenue for the data diversity problem with a clear observation driving their decisions.

Clarity: It its unclear how exactly the algorithms are translated to sample programs.

**Strength And Weaknesses:**

Strengths:

* Intuitive observation and solution to a real world auto-tuning problem.
* Strong results on reduction of auto-tuning time.

Weaknesses

* It is unclear how the sampling process happens. Core-set algorithm is mentioned, but how this is adapted to program sampling is not presented.
* Implementation details of the core algorithm is unclear in the program setting

**Summary Of The Paper:**

The paper introduces an active learning solution to sample configuration points to learn a data-driven cost model. They observe that most configurations are low performant and use a biased program selection strategy to find programs with higher performance. The results show that BALTO can achieve similar or better performance by requiring on a fraction of hardware measurements when the cost model is trained using their sampled dataset.

**Summary Of The Review:**

I enjoyed reading the paper. They clearly articulate a valid observation that program auto-tuning practitioners face in training cost models, which is the training distribution is skewed towards non-performant programs. Therefore, cost models tend to be erroneous in estimating cost of high performant regions, minimizing their efficacy.

Using active learning to come up with the biased sampling strategy IMO is a novel technique to tackle this problem. However, the sampling process that solves eq. 2 and 3 is not clear. The paper mentions about Core-set, however adapting it to programs in not explained properly. I think this is an important detail which should be elaborate more in the paper. I am not sure whether the programs are still generated using a probabilistic grammar or using some other methodology.

The results seem convincing, insofar as the auto-tuning time is reduced drastically. This is because of the efficacy of the cost model.
Overall, I think the paper presents a nice solution to the data imbalance problem, but details were missing to fully appreciate it. I will change my score once the authors provide more clarifying details about their program selection / sampling process.

---

> ### Author Response · Authors · 2022-11-16
> **Response to 3pkd**
>
> We thank reviewer-3pkd for your appreciation and constructive suggestions. We provide a detailed response below.
>
> **Q1: It is unclear how the sampling process happens. Core-set algorithm is mentioned, but how this is adapted to program sampling is not presented.**
>
> A1: To make the program sampling process clearer, we have added a new subsection (i.e., Section 3.2). The program sampling corresponds to step 1 of Figure 3 and is used for generating unmeasured program datasets. BALTO generates unlabelled program datasets by randomly sampling transformed tensor programs from pre-defined optimization tasks. An optimization task corresponds to a machine learning operator (e.g., MatMul) with a specific shape, e.g., (1024, 1024, 1024). For a given task, programs can be randomly generated either by performing random transformations on program sketches (e.g., TenSet and Ansor [1]) or by sampling from probabilistic-language-defined stochastic search spaces (e.g., MetaScheduler [2]). BALTO experiments on both of these two sampling approaches.
>
> The core-set-style algorithm is used to select programs from the unmeasured program dataset for performance measurement. The Core-set selection formulates sample selection as solving the k-center optimization problem of $\min \limits_{S_t \subseteq D_u} \max \limits_{x_i \in S_t} \min \limits_{x_j \in D_l \cup S_t} d(x_i, x_j)\quad s.t. \quad |S_t| \le B_{t}$ where $S_t$ is the samples selected at time step $t$, $d(x_i,x_j)$ is a measure of the distance between sample $x_i$ and $x_j$, and $B_{t}$ is the total budget of samples at step $t$. Since the problem is NP-Hard, a common approach leverages the greedy strategy shown in Algorithm 1 to obtain a 2-OPT solution. BALTO selects programs for measurement based on the core-set selection framework. Firstly, we show an effective way to balance the distribution is to explore the diversity of the predicted performance by setting $d_{out}(x_{i}, x_{j})= |f(x_{i}) - f(x_{j})|$. Then, we show that such a selection approach can suffer from an overestimation problem. To solve this problem, we should encourage the selection to put more weight on programs with high scores on predicted performance. Thus we formulate the biased selection as solving a weighted k-center problem: $\min \limits_{S_{t} \subseteq D_u} \max \limits_{x_{i} \in S_{t}} f(x_{i})  \min \limits_{x_{j} \in D_l \cup S_{t}} d_{out}(x_{i}, x_{j})\quad s.t. \quad |S_{t}| \le B_{t}$. Since the problem is NP-hard, we solve it via a greedy strategy as in Algorithm 2. Finally, we leverage a KL-divergence two decide whether we should apply the biased-diversity-based selection for a given optimization task in a given iteration.
>
> [1] Lianmin Zheng, Chengfan Jia, Minmin Sun, Zhao Wu, Cody Hao Yu, Ameer Haj-Ali, Yida Wang, Jun Yang, Danyang Zhuo, Koushik Sen, et al. Ansor: Generating {High-Performance} tensor programs for deep learning. In 14th USENIX symposium on operating systems design and implementation (OSDI 20), pp. 863–879, 2020a.
>
> [2] Junru Shao, Xiyou Zhou, Siyuan Feng, Bohan Hou, Ruihang Lai, Hongyi Jin, Wuwei Lin, Masahiro Masuda, Cody Hao Yu, and Tianqi Chen. Tensor program optimization with probabilistic programs. ArXiv, abs/2205.13603, 2022.
>
>
> **Q2: Implementation details of the core algorithm is unclear in the program setting.**
>
> A2: We have revised Section 3 for describing the details in the program setting.
> In Section 3.1, we show that BALTO iteratively trains the model via four steps including program sampling, model training, program selection, and performance evaluation. Regarding step-1, BALTO samples programs randomly to form a large unlabelled data set $D_u$ and a small labeled data set $D_l$. Programs in labeled data set are measured on the hardware to obtain the labels. Then BALTO iterates step-2, step-3, and step-4 to train the model based on active learning. Regarding step-2, BALTO trains the model based on $D_l$. Regarding step-3, BALTO leverages the trained model and the labeled data set to select a batch of programs from $D_u$ for evaluation. Regarding step-4, the performances of programs are measured for updating $D_l$. Finally, the model training part outputs a trained model for later program optimization tasks once the total measurement budget is met.
> Section 3.2 and Section 3.3 introduce the program sampling and program selection. These are described in our answer to question 1.
>
> We hope that our response has mostly addressed the reviewer’s concerns. We are happy to continue a discussion to address any other questions the reviewer may have.

---

> > ### Comment · Reviewer_3pkd · 2022-12-12
> > **Post Author-response**
> >
> > I have read the author response. The new details in the paper makes it easier to read. As a domain expert, the paper is approachable and I value its contribution. My score remains the same and I am still positive about the paper.

---

### Official Review · Reviewer_UWiq · 2022-10-25

**Confidence:** 4
**Correctness:** 3
**Technical Novelty And Significance:** 2
**Empirical Novelty And Significance:** 2
**Recommendation:** 6

**Clarity, Quality, Novelty And Reproducibility:**

The paper is well written with enough background information.
I believe most of the experiments can be reproduced.

**Strength And Weaknesses:**

Strength:
- The application of core-set with biased selection is novel. The method is well-motivated and addresses the right problems of TPO.
- The results are impressive. It reduces the required number of data samples by 20x.

Weaknesses:
- Table 1 and Table 2 should include results from more baselines. Currently, they only list BALTO and a simple baseline without any active learning. Although Sec 4.3 does some ablation study, there is a concern about cherry-picking with a single data point in Sec 4.3.
- More analysis of algorithm 1 should be included such as complexity and running time. In the final evaluation, the search time should also be included.
- The real technical contribution of this paper is sec 3.3. But I think the contribution is too small for an ICLR paper.

**Summary Of The Paper:**

The paper proposes BALTO, a biased-diversity-based activation learning approach for fast tensor program optimization. BALTO combines active learning and a biased-diversity-based diversity scheme.
BALTO can achieve the same or higher model accuracy with only 5% of the data samples.

**Summary Of The Review:**

In summary, this paper proposes an effective approach to accelerate tensor program optimization. However, the novelty is limited. I think this is a borderline paper.

---

> ### Author Response · Authors · 2022-11-16
> **Response to UWiq : PART 2**
>
>
> **Q2: More analysis of algorithm 1 should be included such as complexity and running time. In the final evaluation, the search time should also be included.**
>
> A2: We have added the related analysis in Section 3.3. Concretely, the complexity of the biased-diversity-based selection is $O(BM^2)$, where $B$ represents the total measurement budget and $M$ represents the number of programs generated by random program sampling (in Section 3.3) for a given optimization task. Take Tenset for example, the $B$ is 429,810, the $M$ is 4,000, and the total selection time
> is 10.6 minutes. When optimizing the networks, the optimization time of all the baselines is similar since they use the same amount of measurement trials. Regarding BALTO, the total search time is 151 minutes for tuning all the five networks.
>
>
> **Q3: The real technical contribution of this paper is sec 3.3. But I think the contribution is too small for an ICLR paper.**
>
> A3: Reducing the optimization time of TPO is one of the important and challenging problems in ML/DL community.
> To accelerate TPO, the pre-trained ML models can be carefully leveraged to replace a substantial part of the time-consuming hardware measurements with performance predictions.
> However, existing approaches still suffer from the prohibitively expensive program sampling (i.e., more than 4000 GPU hours are required to train a cost model for a specific GPU platform.) for the pre-training of the models.
>
> To address this problem, we propose a brand-new TPO approach BALTO, which is the first work of introducing active learning to reduce the training cost of pre-trained-model-based TPO, to the best of our knowledge.
> In this paper, we make three main contributions.
> First, we conduct a set of experiments to more closely investigate the distribution of large-scale datasets sampled randomly by existing approaches.
> We observe that the random sampling adopted in existing approaches can result in an imbalanced dataset and thus suffers from a heavy redundancy in the training data.
> Such an imbalance problem has been ignored by prior works and thus is never considered in their designs.
> Second, to solve the imbalance problem, we propose the output-diversity-based selection strategy to increase the diversity of program performances.
> Applying the selection strategy directly can lead to an overestimation problem, which can be efficiently addressed further by the proposed biased-diversity-based selection strategy specially designed for BALTO.
> Third, we evaluate BALTO on six hardware platforms, showing that BALTO achieves same or higher model accuracy with a 20$\times$ reduction in performance measurements.
> More importantly, the optimized programs offered by BALTO even outperform that of
> TenSet by 1.07\% on NVIDIA T4.
> Also, to demonstrate the generic ability, we further integrate BALTO into the MetaScheduler to perform auto-tensorization.
> All the aforementioned discussions demonstrate that BALTO can be an important contribution for ICLR.
>
> We hope that our response has mostly addressed the reviewer’s concerns. We are happy to continue a discussion to address any other questions the reviewer may have.

---

> ### Author Response · Authors · 2022-11-16
> **Response to UWiq : PART 1**
>
> We thank reviewer-UWiq for the constructive comments and suggestions. We provide a detailed response below.
>
> **Q1: Table 1 and Table 2 should include results from more baselines. Currently, they only list BALTO and a simple baseline without any active learning. Although Sec 4.3 does some ablation study, there is a concern about cherry-picking with a single data point in Sec 4.3.**
>
> A1: To address this concern, we add additional experiments on five active learning baselines including GSx, GSy, iGS, ALT, and Ours1. Among the five baselines, the first four baselines are representative active learning approaches for regression problems and the last one is our proposed output-diversity-based active learning approach which is based on the core-set selection.
> Concretely, the four AL baselines include:
> 1) GSx [1] selects programs by maximizing the feature diversity  greedily;
> 2) GSy [2] selects programs by maximizing the label diversity greedily;
> 3) iGS [2] selects programs by maximizing the label and feature diversity  greedily;
> 4) ALT [3] selects programs by the uncertainty of the model prediction.
>
> Each result is averaged on five independent experiments with a 95\% confidence interval. The experimental results in Table 1 show that BALTO outperforms other active learning approaches with higher prediction accuracy under the same amount of measurement requirements. The experimental results in Table 2 show that other AL-based approaches can degrade the compiler's optimization ability due to lower prediction accuracy.
>
> The new Table 1 is as follows:
> |XGBoost|GPU-1|GPU-2|CPU-1|CPU-2|CPU-3|CPU-4|
> |-------|----|----|----|----|----|----|
> |TenSet|$84.7\pm0.4$|$84.9\pm0.1$|$82.7\pm0$|$83.0\pm0$|$83.9\pm0$|$79.2\pm0$|
> |GSx|$81\pm0$|$80.9\pm0$|$78.1\pm0.27$|$78\pm0$|$79.6\pm1.01$|$75.6\pm0$|
> |GSy|$81.3\pm0$|$79.8\pm0$|$78.8\pm0$|$80\pm0$|$81.4\pm0$|$76.5\pm0$|
> |iGS|$82.3\pm0$|$80.9\pm0$|$79\pm0$|$80.1\pm0.2$|$81.4\pm0.3$|$76.8\pm0.7$|
> |ALT|$85.3\pm0$|$85.2\pm0$|$81.5\pm0$|$83.7\pm0$|$84.6\pm0$|$78.8\pm0$|
> |**Ours1**|$87.3\pm0.5$|$87.6\pm0$|$86.2\pm0$|$85.6\pm0$|$87.3\pm0.1$|$80.6\pm0$|
> |**Ours2**|**88.8**$\pm0.2$|**88.9**$\pm0.1$|**87.3**$\pm0.3$|**86.4**$\pm0.5$|**87.9**$\pm0.2$|**81.6**$\pm0.1$|
> |**MLP**|**GPU-1**|**GPU-2**|**CPU-1**|**CPU-2**|**CPU-3**|**CPU-4**|
> |TenSet|$90.5\pm0.3$|$90.2\pm0.5$|$86.5\pm0.8$|$86.1\pm0.4$|$88.1\pm0.5$|$80.8\pm0.8$|
> |GSx|$83.5\pm1.6$|$83.3\pm0.9$|$76.2\pm0.2$|$74.8\pm0.3$|$77.6\pm1.7$|$69.5\pm0.7$|
> |GSy|$82.9\pm1.9$|$82.2\pm0.3$|$75.1\pm1.1$|$72.9\pm1.6$|$74.7\pm0.4$|$68.9\pm1.5$|
> |iGS|$82.3\pm0.4$|$81.4\pm0.2$|$74.1\pm2.2$|$73.9\pm2$|$75.3\pm0.9$|$69.2\pm2.4$|
> |ALT|$86\pm0.4$|$84.3\pm0.2$|$77.3\pm1.2$|$79.2\pm1.2$|$75.3\pm0.9$|$69.2\pm2.4$|
> |**Ours1**|$89.6\pm0.9$|$89.8\pm0.7$|$86\pm0.4$|$86.4\pm0.8$|$88.1\pm0.6$|$79\pm1.1$|
> |**Ours2**|$90.4\pm0.6$|$90.3\pm0.6$|$86.9\pm1.0$|**87.2**$\pm1.1$|**89.1**$\pm1.3$|$81.1\pm1.1$|
>
> The new Table 2 is as follows:
> |XGBoost|ResNet-50|MobileNet-v2|ResNext-50|BERT-tiny|BERT-base|
> |-------|----|----|----|----|----|
> |TenSet|$3.89\pm0.4$ms|$0.60\pm0.04$ms|$3.41\pm0.31$ms|$3.06\pm0.30$ms|$10.8\pm0.2$ms|
> |GSx|$3.92\pm0.1$ms|$0.62\pm0.08$ms|$3.62\pm0.09$ms|$3.11\pm0.05$ms|$12.2\pm0.7$ms|
> |GSy|$4.39\pm0.4$ms|$0.63\pm0.05$ms|$3.65\pm0.56$ms|$3.03\pm1.10$ms|$11.1\pm1.1$ms|
> |iGS|$3.81\pm0.1$ms|$0.65\pm0.03$ms|$3.71\pm0.73$ms|$3.26\pm0.21$ms|$11.0\pm1.0$ms|
> |ALT|$3.62\pm0.1$ms|$0.59\pm0.02$ms|$3.30\pm0.04$ms|$2.81\pm0.13$ms|$11.1\pm1.6$ms|
> |**Ours1**|$3.68\pm0.5$ms|$0.59\pm0.06$ms|$3.27\pm0.1$ms|$2.84\pm0.09$ms|$10.9\pm0.3$ms|
> |**Ours2**|**3.30**$\pm0.2$ms|$0.58\pm0.04$ms|**3.23**$\pm0.17$ms|**2.76**$\pm0.05$ms|$10.8\pm0.4$ms|
> |**MLP**|**ResNet-50**|**MobileNet-v2**|**ResNext-50**|**BERT-tiny**|**BERT-base**|
> |TenSet|$3.28\pm0.15$ms|$0.57\pm0.03$ms|$3.26\pm0.32$ms|$2.80\pm0.09$ms|$10.0\pm0.4$ms|
> |GSx|$4.13\pm0.17$ms|$0.64\pm0.08$ms|$3.50\pm0.32$ms|$3.16\pm0.13$ms|$11.5\pm0.8$ms|
> |GSy|$3.79\pm0.58$ms|$0.62\pm0.07$ms|$3.40\pm0.24$ms|$2.92\pm0.15$ms|$11.5\pm0.5$ms|
> |iGS|$3.58\pm0.09$ms|$0.62\pm0.06$ms|$3.43\pm0.13$ms|$2.91\pm0.01$ms|$11.3\pm1.1$ms|
> |ALT|$4.04\pm0.25$ms|$0.75\pm0.19$ms|$3.30\pm0.16$ms|$3.06\pm0.16$ms|$11.6\pm0.9$ms|
> |**Ours1**|$3.43\pm0.42$ms|$0.60\pm0.02$ms|$3.30\pm0.33$ms|$2.79\pm0.07$ms|$10.8\pm0.5$ms|
> |**Ours2**|$3.33\pm0.17$ms|$0.58\pm0.00$ms|$3.14\pm0.17$ms|$2.71\pm0.13$ms|$9.9\pm0.4$ms|
>
> [1] Hwanjo Yu and Sungchul Kim. Passive sampling for regression. 2010 IEEE International Conference on Data Mining, pp. 1151–1156, 2010.
>
> [2] Dongrui Wu, Chin-Teng Lin, and Jian Huang. Active learning for regression using greedy sampling.
>
> [3] Xi Zeng, Tian Zhi, Zidong Du, Qi Guo, Ninghui Sun, and Yunji Chen. Alt: Optimizing tensor compilation in deep learning compilers with active learning. In 2020 IEEE 38th International Conference on Computer Design (ICCD), pp. 623–630. IEEE, 2020.

---

> ### Comment · Reviewer_UWiq · 2022-12-03
> **Raise my score after reading rebuttal**
>
> After reading the rebuttal, I think some of my concerns are addressed. I raise my score to 6

---

### Decision · Program_Chairs · 2023-01-20

**Decision:**

Accept: poster

**Justification For Why Not Higher Score:**

While the paper has many positive aspects, questions related to novelty and clarity of writing still remain, preventing me from recommending a higher score.

**Justification For Why Not Lower Score:**

Two of the reviewers were domain experts working in this exact application area (program optimization), and they were the most positive about the paper. I feel that even though the paper does not break novel ground from the ML techniques standpoint, the fact that the approach seems to be genuinely useful in applications makes it a worthy candidate for acceptance.

**Metareview: Summary, Strengths And Weaknesses:**

The paper proposes an approach called biased active-learning for tensor program optimization, or BALTO. The high level idea is to use diversity-based active learning for selecting program samples (which increases the proportion of high-performance programs in the training set). This mitigates a known failure mode in TPO, which rely on random sampling and therefore are heavily imbalanced towards low-performance programs. Experiments on six different hardware platforms show the benefits of BALTO with up to 20X reduction in measurement costs.

Original reviews on this paper were split. Some reviewers were positive about the paper, highlighting that the authors had addressed a real-world application with measurable improvements over current state-of-the-art. Other reviewers pointed out issues with the clarity of writing, as well as the (relative) lack of novelty, given that other approaches for TPO based on active learning have already been proposed. Additional baseline comparisons, and more details about the experimental setup, may have helped here.

After a discussion among the reviewers and the AC, the overall majority sentiment seemed to point favorably in the direction of the paper. The authors are recommended to take into account the comments made by the reviewers while preparing revisions.

Recommendation: accept.


**Note From Pc:**

if the above contains the word "oral" or "spotlight" please see: "oral" presentation means -> notable-top-5% and "spotlight" means -> notable-top-25%. As stated in our emails, we are disassociating presentation type from AC recommendations

**Summary Of Ac-Reviewer Meeting:**

On the plus side:
- One reviewer felt that they felt well connected with the paper; the authors had addressed a real problem faced in program optimization (which they had faced during their own research), and the solution was satisfactory. The way the authors had presented the methodology was fuzzy, but in the response phase the authors had made an effort to give more specifics.
- Another reviewer concurred: it is indeed true that most training samples in TPO datasets are not very helpful, so the proposed solution (diversity-based active learning) made a lot of sense. While the paper did have weaknesses (mainly in the experimental evaluation and writing), the insights in the paper were valuable to the ML systems/compilers community, and the real-world impact -- e.g. ability to improve optimization ability matmul/2D convolution -- made it worth shepherding the paper through.

On the minus side:
- Concerns remained about the novelty of the proposed techniques (particularly since there are other active learning-based TPO approaches already) and the writing (both in terms of clarity as well as accessibility of the writing).
- There was also questions as to whether the paper could be sufficiently modified at this phase to address both these issues.

The reviewers were requested to revisit their scores in light of the discussion and the satisfactoriness of the authors' responses.